UMEDNet: a multimodal approach for emotion detection in the Urdu language

Majeed Adil adilmajeed350@gmail.com
Mujtaba Hasan
School of Computing, National University of Computer and Emerging Sciences, Islamabad , Islamabad, Punjab , Pakistan
Comai Sara
Electronic publication date: 2025 May 1
Publication date: 2025
Volume: 11
Electronic Location ID: e2861
Received 2024 Oct 25; Accepted 2025 Apr 4
Copyright: © 2025 Majeed and Mujtaba
Copyright year: 2025
Copyright holder: Majeed and Mujtaba
License: This is an open access article distributed under the terms of the Creative Commons Attribution License, which permits unrestricted use, distribution, reproduction and adaptation in any medium and for any purpose provided that it is properly attributed. For attribution, the original author(s), title, publication source (PeerJ Computer Science) and either DOI or URL of the article must be cited.
License URL: https://creativecommons.org/licenses/by/4.0/

Keywords: Urdu, Multimodal, Emotion detection, Urdu corpus, Classification

Funding: The authors received no funding for this work.

==============================
Emotion detection is a critical component of interaction between human and computer systems, more especially affective computing, and health screening. Integrating video, speech, and text information provides better coverage of the basic and derived affective states with improved estimation of verbal and non-verbal behavior. However, there is a lack of systematic preferences and models for the detection of emotions in low-resource languages such as Urdu. To this effect, we propose Urdu Multimodal Emotion Detection Network (UMEDNet), a new emotion detection model for Urdu that works with video, speech, and text inputs for a better understanding of emotion. To support our proposed UMEDNet, we created the Urdu Multimodal Emotion Detection (UMED) corpus, which is a seventeen-hour annotated corpus of five basic emotions. To the best of our knowledge, the current study provides the first corpus for detecting emotion in the context of multimodal emotion detection for the Urdu language and is extensible for extended research. UMEDNet leverages state-of-the-art techniques for feature extraction across modalities; for extracting facial features from video, both Multi-task Cascaded Convolutional Networks (MTCNN) and FaceNet were used with fine-tuned Wav2Vec2 for speech features and XLM-Roberta for text. These features are then projected into common latent spaces to enable the effective fusion of multimodal data and to enhance the accuracy of emotion prediction. The model demonstrates strong performance, achieving an overall accuracy of 85.27%, while precision, recall, and F1 scores, are all approximately equivalent. In the end, we analyzed the impact of UMEDNet and found that our model integrates data on different modalities and leads to better performance.

Introduction

Emotions are an intrinsic part of human nature and part of our daily life (Baltrušaitis, Ahuja & Morency, 2018; Abdullah et al., 2021). One of artificial intelligence (AI)’s long-standing goals has been to build systems capable of detecting and interpreting emotional cues. Interpersonal relationships, knowledge, insight, and perception, among other things, depend on emotion (Cevher, Zepf & Klinger, 2019; Dadebayev, Goh & Tan, 2022). Since emotion acquisition and experiences are highly crucial for inter-communication in social situations, it has become important for human daily functioning to understand emotions. Emotion detection is a critical field of research in human-computer interactions that has seen an increasing demand for automated emotion detection systems (Ahmed, Al Aghbari & Girija, 2023). Research focused on emotion recognition has utilized voice, text, face cues, and electroencephalogram (EEG)-based brain waves. However, given the growth and expansion of human-computer interaction, emotion detection systems have become a method for developing pleasant or intuitive user interaction (Priyasad et al., 2020). In 2020, the emotion detection and recognition market was valued at USD 19.87 million, and is projected to reach USD 52.86 million by 2026, expanding at a compound annual growth rate (CAGR) of 18.01 percent during 2021 to 2026 (Ahmed, Al Aghbari & Girija, 2023).

Emotion detection has become interesting due to its applications. We believe emotion detection is important and interesting for various purposes such as business development and improving the quality of products (Soleymani, Pantic & Pun, 2011; Patwardhan, 2017). For example, emotion is more important than price and functionality for selling or purchasing a product (Alm, 2012). Here, we can use emotion detection to sell products. To increase the sale of the product, it detects, analyzes, and predicts consumer emotions based on happiness or sadness as feedback for the product (Gaind, Syal & Padgalwar, 2019). Emotion detection is important because it can be predicted to increase the performance of conversations and train machines to respond according to a person’s emotions. People now openly express their opinions, thoughts, comments, and feedback on a specific object, product, commodity, political event, and other viral news (Majeed et al., 2022). Humans may communicate their emotions through gestures, facial expressions, speech, and text. In recent years, many researchers have attempted to identify human emotion from these expressions (Priyasad et al., 2020; Lee, Kim & Cheong, 2020; Krishna & Patil, 2020). For instance, there are multiple channels of emotional communication, e.g., audio, video, and text. The data from these channels require multimodality, which refers to combining different modes of data to extract the required information, to handle all of it simultaneously. Therefore, there is a clear emphasis on the relevance and importance of multisensory integration in processing human emotions.

Multimodal analysis of emotions is becoming an increasingly popular field of study, because it offers a more comprehensive and accurate way to understand and analyze emotions. The traditional, language-based definition of sentiment analysis has been expanded to include other relevant modes. To achieve this, multiple methods and techniques are combined. Frequently, these methods are based on massive data sets, semantic rules, and machine learning (Mittal et al., 2020). Finding a person’s emotions through a series of images (video) defines a continuous and real process, enhancing the integrity of emotions discovered. Typically, a video contains audio and video data, which can be viewed as distinct sources such as audio, images, and text that can be analyzed in various ways (Abdullah et al., 2021). For example, the content of the speech and the tone of the voice can be analyzed from the audio. Facial expressions, posture, and the surrounding environment can be used to infer emotions from images.

Several studies have been conducted on emotion detection. However, only a small number of studies have been published in languages other than English across the globe. More than 100 million people worldwide use Urdu, the national language of Pakistan, as their primary language. Urdu is the ninth most spoken language in the world, based on the number of native speakers (Bashir et al., 2023). It is written from right to left in an enhanced variant of the Perso-Arabic script. A large proportion of Urdu vocabulary is derived from other languages. Urdu morphology is derived from Arabic, Persian, and South Asian grammar (Anwar, Wang & Wang, 2006). It is typically written in Nastalique, a context-sensitive, exceedingly complex, cursive script. To achieve the objective, we require a corpus and multimodal architecture for comparing, evaluating, and analyzing emotion detection systems as shown in Fig. 1. To reach a solution, these are some of our contributions to emotion detection:

Figure 1 Abstract diagram: this diagram demonstrates that we have input in several modalities; namely text, voice, and visual.

After processing the raw information, our objective is to extract emotions from the inputs given in Urdu.

1. Collection and manual annotation of the corpus (Urdu Multimodal Emotion Detection (UMED) corpus) for multimodal (visual, audio, and text) emotion detection for the Urdu language.

2. UMED corpus deals with five common emotions of happiness, love, sadness, anger, and neutrality for the Urdu language.

3. After annotation, to ensure the diversity and quality of the corpus, we apply inter-annotator agreement and Cohen’s Kappa statistics score.

4. Development of a novel classifier for emotion detection in the Urdu language.

5. Application of a novel approach/architecture for multimodal classification of emotion detection.

The article is organized as follows: First, existing literature and limitations of current multimodal emotion detection methods are reviewed. Following this, the proposed approaches for multimodal emotion detection are provided. Next, we describe the experiments and their evaluations. Finally, we conclude the article.

Related works

In the current era of smartphones and social networking sites, many individuals express their ideas, emotions, and successes using audio and video files because of their inherent characteristics. We have studied several articles that directly or indirectly address multimodal emotional detection in English, Arabic, and German. However, there is relatively little literature on Urdu. Researchers employ a variety of methods to identify emotions. In this literature, the most prevalent strategy has been explored. This section focuses on emotion detection using multimodality and contains recent work in the field.

Audio and text modality in emotion detection

Siriwardhana et al. (2020) investigate the feasibility of using modality-specific “bidirectional encoder representations from transformers (BERT)-like” pre-trained self-supervised learning (SSL) architectures to represent voice and text modalities for multimodal speech emotion identification. They demonstrate experimentally that jointly fine-tuning “BERT-like” SSL designs achieves state-of-the-art outcomes on three publicly accessible datasets (IEMOCAP, CMU-MOSEI, and CMU-MOSI). The authors also compare two approaches to merging voice and text modalities, demonstrating that a simple fusion mechanism can outperform more complicated ones when employing SSL models. Priyasad et al. (2020) presents an attention-based multi-modal emotion recognition model based on both acoustic and textual data. A SincNet layer hierarchically structured with parameterized sinc functions and band-pass filters is used to extract the auditory features that are fed into a deep convolutional neural network (DCNN). This approach learns filter banks optimized for emotion recognition. The regarded experiments prove that the proposed system is superior to existing approaches.

The main goal of Hazarika et al.’s (2018a) feature-level fusion method is to improve the integration and interaction of the multiple inputs with the proposed self-attention-based framework. They perform a noise stability test to once again demonstrate performance in noisy conditions. These outcomes endorse the optimistic hypothesis introduced in this method and present it as being a drastic improvement within the area of feature-level fusion and its applicability to more extensive multimodal issues. Padi et al. (2022) incorporated a neural network system for emotion recognition that is based on a late fusion of transfer-learned and fine-tuned models of speech and text data. In the case of speech signals, they fine-tune a residual network (ResNet) based model that is trained initially on a large-scale speaker recognition task via transfer learning. They also apply a spectrogram augmentation technique to recognize emotions from speech. For text, they use a fine-tuned BERT for representing and recognizing emotions. Additionally, the proposed system performs the fusion of the ResNet and BERT-based model scores employing a late fusion approach to enhance the performance.

Video and audio modality in emotion detection

Abu Shaqra, Duwairi & Al-Ayyoub (2023) provides a multimodal system for identifying and labeling emotional expressions in an Arabic-based dataset. They first assessed the usefulness of audio and video data separately and later revealed what happened when they were combined into a single model. Emotion identification algorithms that rely on audio data show considerable performance improvements when speaker gender is known in advance. By contrasting audio-based and visual-based systems, they see that each model excels at recognizing a different set of emotional labels. Chen et al. (2014) conducted an exploratory investigation by implementing state-of-the-art emotion recognition techniques in the analysis of interview videos. They developed a video corpus by inviting actors to read three types of scripts related to call center work composure and employee competencies. This corpus was used to assess the ability of Emotient’s FACET toolkit to identify emotions and evaluate the actors’ performances. Additionally, they utilized the Linear Discriminant Classifier (LDC) emotional speech corpus and several open-source tools to construct speech emotion classifiers for four emotion categories.

Chong, Jin & He (2019) proposes EmoChat, an online chatting system that adopts an innovative feature to tag users’ emotions during a chat conversation without being prompted. EmoChat utilizes both facial expressions and text-based inputs to evaluate users’ emotions comprehensively. To fuse these two complementary modalities, they present an information entropy-based method. To increase the accuracy of recognizing emotions they use the Hidden Markov Model (HMM) which takes into account contextual information. Mamieva et al. (2023) presents a new multimodal emotion recognition system using both facial expression and speech data utilizing attention-based frameworks. This technique compares the most relevant features of the facial and spoken parts extracted using different encoders. This strategy does not have the weaknesses of uni-modal systems because it uses additional information from the other modality and improves the recognition of emotions. These modalities are integrated by an architecture called a fusion network to generate a multimodal feature vector which is followed by a classification layer to identify the affective state resulting from the computation.

Tzirakis et al. (2017) introduces an emotion recognition system based on auditory and visual modalities. They employ a convolutional neural network (CNN) to extract features from the speech. For the visual modality, a deep residual network (DRNN) is used. The experiments on the uni-modal modality also indicate that proposed models yield substantially higher accuracy on the test set as compared to the other models accessing the RECOLA database including those introduced to the AVEC2016 challenge, thus proving the effectiveness of learning features that are more suitable for the task under consideration. Moreover, they have realized that their multimodal model outperforms others, not only in terms of valence but also in terms of arousal.

Video, audio, and text modalities in emotion detection

Using multimodal feature extraction and hierarchical modeling, Hazarika et al. (2018b) proposed the Interactive Conversational Memory Network (ICON), a framework for analyzing videotaped conversations that can detect multiple emotions. Using the memories made this way, they can predict how someone will feel in a segment of the video conversation. Lai, Chen & Wu (2020) presents a model for distinguishing across contexts using different contextual window sizes based on recurrent neural networks (DCWS-RNNs). The model employs four RNNs, each using a unique contextual window size. These window widths can represent various contexts’ implicit weights. In addition, four RNNs are used to autonomously model the different situations into memory. Then, these recollections and the test segments are combined using attention-based multiple hops.

Pan et al. (2020) investigated a hybrid fusion method known as a multimodal attention network (MMAN). They propose Convolutional Long Short-Term Memory with Multi-Modal Attention (cLSTM-MMA), a novel multimodal focus mechanism that encourages attention across three modalities and selectively integrates information. Late fusion involves combining cLSTM-MMA with other uni-modal sub-networks. The results show that using visual and textual clues greatly enhances the accuracy of emotion recognition in spoken language. Cevher, Zepf & Klinger (2019) present an in-car experiment they conducted to test emotion identification from spoken interactions across three different modalities. They employed commercially available audio and facial emotion identification technologies. They compared them to a neural transfer learning strategy for emotion recognition from text, which used resources from other domains. Using transfer learning makes it possible for models built using data from different fields to succeed.

Schmitz, Ahmed & Cao (2022) work with audio, text, and video data types and their combinations. In this case, text only performs worst in bias metrics yet contributes significantly to the models scores, questioning the need for multimodal solutions if one is seeking both fairness and high accuracy at the same time. Audio has slightly better results in terms of accuracy and fairness compared to video whose performance is very poor. The combination of audio and video stands out from the other bimodal models while incorporating video decreases both predictiveness and fairness for trimodal models. Caschera, Grifoni & Ferri (2022) proposes a method which generalizes the concept of multimodal data based on the sequence of features derived from facial expressions, voice, gestures, and natural language processing in linguistic terms. In this article, each of the emotions was modeled as an HMM, and these were trained and tested using multimodal samples containing seven basic emotions. The classification rates of the emotions were studied and it was observed that the experiment yielded good results.

Illendula & Sheth (2019) utilizes the significance of various modalities derived from social media posts for the task of emotion classification with the help of advanced deep learning architectures. They have described a multimodal emotion classification method utilizing emojis, textual, and visual elements. They have shown that the three modalities convey different information to produce emotions.

To summarize and overview existing research are shown in Table 1. In overview, there have been some studies on the effectiveness of multimodal approaches for emotion detection, and the results have been mixed. Some studies have found that multimodal approaches can improve the accuracy of emotion detection, while others have found little or no improvement. Developing evaluation resources constitutes a crucial difficulty within this discipline. Development of natural language processing (NLP) tasks depends on benchmark corpora for conducting both development and evaluation. Emotion detection corpora exist for languages like English and French but there are no such resources available for South Asian languages specifically Urdu. The domain of emotion detection in Urdu lacks any publicly accessible dataset at present. The lack of standard Urdu language resources requires us to develop a corpus first which we will utilize in constructing our multimodal emotion detection framework.

Table 1 Overview of prior research on multimodal emotion detection.

Article	Modality	Language	Model	Contribution	Evaluation matrix	
Hazarika et al. (2018a)	Audio and text	English	Self-attention based multimodal	A new feature-level fusion method based on self-attention mechanism.	Accuracy 72.1%, F172.2%, UAR 72.1%	
Siriwardhana et al. (2020)	Audio and text	English	Pre-Trained SSL models	Jointly fine-tuned modality-specific SSL models that represent speech and text.	Accuracy 88.2%, F1 88.5%	
Priyasad et al. (2020)	Audio and text	English	DCNN, RNN	Combines acoustic and textual information.	Accuracy 80.51%	
Padi et al. (2022)	Audio and text	English	ResNet	Presented a multimodal emotion recognition framework using transfer learning.	UA 75.97%, WA 75.01%	
Chen et al. (2014)	Audio and video	English	FACET	Create a corpus and perform an evaluation of Emotient’s FACET toolkit.	Spearman Correlation	
Tzirakis et al. (2017)	Audio and video	English	CNN and Deep Residual Network	Pretrained the speech and visual networks separately to speed up the model’s training.	Arousal 0.788, Valence 0.732	
Chong, Jin & He (2019)	Audio and video	English	HMM	Design a confidence-aware method to fuse facial expression and text data, and collect a multimodal chat dataset.	Accuracy 76.25%	
Mamieva et al. (2023)	Audio and video	English	CNN	The time and spectral information were used to avoid losing important information.	Accuracy 80.7%, F1 73.7%	
Abu Shaqra, Duwairi & Al-Ayyoub (2023)	Audio and video	Arabic	MLP, CNN-LSTM	Three schemes are proposed in this work for both detection and recognition tasks.	Accuracy 75%	
Hazarika et al. (2018b)	Audio, video and text	English	ICON	It introduced a multimodal approach that provides comprehensive features.	Accuracy 64%, F1 63.5%	
Cevher, Zepf & Klinger (2019)	Audio, video and text	German	NN	Introduced transfer learning to adapt models trained in the German language.	F1 76%	
Illendula & Sheth (2019)	Audio, video and text	English	Bi-LSTM	Hypothesize that visual, textual, and emoji features from a social media post all contribute to predicting user emotions.	Accuracy 71.98%, F1 70.78%	
Lai, Chen & Wu (2020)	Audio, video and text	English	DCWS-RNNs	Propose a DCWS-RNNs model to distinguish the contexts of the test utterance and adopt a multimodal approach.	Accuracy 64.5%, F1 62.43%	
Pan et al. (2020)	Audio, video and text	English	cLSTM-MMA	Propose MMAN and fusion method.	Accuracy73.98%	
Schmitz, Ahmed & Cao (2022)	Audio, video and text	English	NN	Analyze and mitigate gender bias in the current state-of-the-art transfer learning algorithms for different modalities including audio, text, and video.	Accuracy 65.19%, F1 65.39%	

In addition, Urdu does not have pre-trained multimodal embeddings and thus it is not easy to develop emotion detection models. To overcome this limitation, we propose a new multimodal architecture incorporating text, speech, and video embeddings. This article aims to present a robust multimodal emotion detection system made up of machine learning and deep learning techniques. When emotion prediction deals with languages with low resources, it faces serious challenges scarcity of publicly available datasets. In contrast to high-resource languages, such as English, there are no such corpora or well-set models for deep learning to effectively train in low-resource languages. In this case, lack of resources impedes progress in the unimodal and multimodal emotion recognition tasks. Given these challenges, our study contributes to the research on low-resource language for emotion recognition by leveraging a multimodal approach. With our approach, we aim to overcome dataset limitations and develop a model for multimodal emotion detection.

Umednet: proposed approach

UMEDNet, an acronym for Urdu Multimodal Emotion Detection Network, is intended to predict emotions from multimodal input of video, voice, and text. Here, we provide an overview of video feature extraction and the methodology of speech and text feature extraction. Lastly, we elaborate on how these features are integrated and applied to estimate emotions in the last phase of the proposed architecture.

Video features extraction

Video feature extraction is important for understanding and analyzing video data to extract important information from frames. For tasks like emotion detection, identity verification, and behavior analysis, faces appearing in videos are important. We focus on extracting features of the facial regions of individuals appearing in videos. To do that, each video is processed frame by frame, extracting facial features from faces detected in the video. We aim to capture high-dimensional representations that capture the spatial and temporal aspects of the video, so video data can be robustly modeled using the emotion detection system. For face detection, the MTCNN (Multi-task Cascaded Convolutional Networks) (Xiang & Zhu, 2017) is used to locate facial regions within each frame efficiently. After face detection, facial embeddings are generated using the FaceNet model (Schroff, Kalenichenko & Philbin, 2015), which is a pre-trained deep learning model. The FaceNet model extracts rich and discriminative features, encoding important facial attributes and expressions essential for accurate emotion detection and other facial analysis tasks. The purpose of MTCNN is to detect faces and align them correctly for additional processing. The face recognition model, FaceNet converts faces into embedding data for identification and verification tasks. The latest DINOv2 (Oquab et al., 2023) model operates as a self-supervised vision transformer that provides generalized image representation learning and delivering optimal outcomes for object recognition and segmentation tasks but does not target face-related processing. MTCNN and FaceNet specialize in facial analysis but DINOv2 is a comprehensive vision model capable of processing different images beyond face data.

Video features processing

A transformer-based model is adopted in this study to evaluate facial embeddings derived from frames in a video for the goal of emotion detection as shown in Fig. 2. The frame processing model commences by feeding input facial embeddings to a higher-dimensional hidden space. This projection is followed by positional encoding which captures temporal information important to analyse video data. The transformer encoder section, which is the core of the whole model, includes multiple layers and attention. The given architecture is good at capturing fine-grained temporal relationships between frames, which makes it possible to interpret facial expressions. The model incorporates padding and masking mechanisms to effectively manage variable-length sequences, ensuring that only valid frames contribute to the output. This makes it possible for the model to incorporate sequenced information from videos which is important for the accuracy of the detected emotion. By including facial embeddings into the transformer architecture, there are lots of advantages in utilizing temporal dynamics, which is beneficial for models that analyze facial expressions in video. After extracting facial embeddings, we combine these embeddings with other modalities to predict the emotion.

Figure 2 The visual modality pipeline: this enables the processing of video frames to obtain facial features for emotion detection.

MTCNN starts the detection of facial landmarks which enables precise face alignment. The high-dimensional facial characteristic extraction process starts after FaceNet extracts the embeddings from faces. A transformer-based model analyzes the facial embeddings and determines how temporal and spatial relationships function between frames to produce enhanced emotion representation.

Audio features extraction

To begin with, audio from the video files is obtained using the MoviePy library. It involves extracting video files and splitting audio streams from the file for analysis. For feature extraction, we fine-tune the Wav2Vec 2.0 model (Babu et al., 2021) which is a self-supervised learning model used for automatic speech recognition (ASR). Designed by Facebook AI researchers, Wav2Vec 2.0 introduces an effective way of improving the ASR since it does not rely on large labeled datasets. The model is pre-trained on unlabeled audio data, where it learns to generate latent speech representations by masking parts of the input and predicting these masked segments. This process is similar to masked language modeling in NLP, enabling the model to perceive phonetically and linguistically relevant features. By fine-tuning Wav2Vec 2.0, we extract the embeddings that define key aspects of the signal while including phonetic and linguistic information. These feature embeddings are used for tasks such as emotion recognition, which are important to create an adequate base for speech-based applications.

To extract meaningful audio embeddings for Urdu, we fine-tuned Wav2Vec2 with an Urdu vocabulary for feature extraction. During fine-tuning, we froze the first six transformer layers and kept pre-trained representations while saving on the computational overhead. Hidden dropout and mask time probability were optimized as key hyperparameters to make training more stable and adapt the model. It was also evaluated on performance results, using the word error rate (WER). Wav2Vec2 may introduce biases for high-resource languages which may negatively affect Urdu performance. This study does not directly address bias, but acknowledges this limitation and suggests future work to explore the mitigation strategies for improved multimodal emotion detection.

The Wav2Vec2 model was selected due to its excellence in speech recognition tasks with language-specific functionality and rich representation capabilities. Wav2Vec2 behaves as a self-supervised model that can effectively learn speech features automatically while using unlabeled data with pretraining multilingual datasets. The model accuracy improves because it undergoes fine-tuning specifically for an Urdu dataset implementation. While other models like Hidden-Unit BERT (HuBert) (Hsu et al., 2021) pivot their attention on the phoneme level, Wav2Vec2 demonstrates better performance in automated speech recognition from end to end. The model provides remarkable performance and efficiency.

Audio features processing

In the architecture, all audio features are passed through a feed-forward network (FFN) as shown in the Fig. 3. In this FFN, we apply two fully connected layers to expand the input dimension of the audio features into a hidden space. They use a linear transformation followed by rectified linear unit (ReLu) activation, introducing non-linearity into the model. To address over-fitting of data, we used dropout regularizations. The last outcome of the network is a feature map of the audio that has been transformed and encoded for combination with other modalities.

Figure 3 The audio modality pipeline utilizes: this pipeline to extract speech-based emotional cues.

The audio signals enter Wave2Vec 2.0 for conversion into speech representations that detect phonetic and prosodic variations. The audio features extracted from the system receive processing through an FFN network for generating discriminative embeddings that help classify emotions from multiple modalities.

Text features extraction

Extracting features from audio data is accomplished by using automatic transcription that transcribes speech into text form. This uses the Whisper (Radford et al., 2023) model which has gained recognition for accurate and effective results in text transcribing. We utilize the large variant of Whisper, to achieve the highest possible quality of transcription. After transcribing the audio into text, we can analyze its linguistic features through natural language processing. During this phase, XLM-RoBERTa (Li, He & Xu, 2021) is used for feature extraction from the whisper transcriptions. The XLM-RoBERTa large model utilizes its ability to improve the quality of word embedding across multiple languages. XLM-RoBERTa is capable of capturing contextual relationships in words, therefore it is useful in several linguistic tasks. In addition, we employ this model to extract relevant information from text. XLM-RoBERTa is a powerful language representation model for capturing contextual information given the distribution of words in the text. The connected Urdu text is passed to the XLM-RoBERTa model which produces feature vectors that accurately represent the semantics of the text. Since Urdu is a low-resource language compared to English or other widely used languages, it should not come as a surprise if this bias affects textual embeddings. Referring to the architectural details, we trained on XLM-Roberta-large with 24 transformer layers, 1,024 dimensions per layer, and 16 heads per layer. To be compatible with the Urdu text processing, we did not modify the context window size. To tokenize and process Urdu transcriptions with padding and truncation to a fixed maximum length.

Text features processing

Text features are further processed through another feed-forward network as shown in Fig. 4. This network has the same structure as the one used for audio, having fully connected layers coupled with ReLU activation and dropout regularization. Textual features are learned in the network and transformed into a hidden state. The FFN generates a fixed-shape vector that can be used for projection into a shared space, similar to other modalities, for emotion detection.

Figure 4 Text modality pipeline: text features are obtained with the help of XLM-R.

XLM-R extracts contextualized word embeddings from texts through which it captures linguistic and semantic relationships at the word level. The FFN receives embeddings and executes a refinement process before creating predictive text-based emotion embeddings.

Multimodal feature projection and emotion prediction

Our model employs projection functions to map the features of various modalities (video, audio, and text) to shared common spaces represented in Fig. 5 to facilitate more effective multimodal alignment and fusion. For every modality M, we define a projection function PM that projects features from its original feature space into a common feature space. This necessary projection function enables the interaction between modalities and improves the overall performance of the emotion detection task.

Figure 5 UMEDNet is a multimodal emotion detection model for the Urdu language that uses text features with XLM-R, facial features with MTCNN and FaceNet, and audio features with Wav2Vec2.

These features are then processed and projected into a shared latent space for emotion prediction.

Let V∈RdV, A∈RdA, and T∈RdT represent the feature vectors from video, audio, and text modalities, respectively, where dV, dA, and dT denote their respective dimensions. To ensure compatibility and facilitate effective fusion, each modality is projected into a shared latent space of dimension d using the following projection functions:

(1) Vproj=PV(V)=f(WVV+bV),WV∈Rd′×dV,bV∈Rd′

(2) Aproj=PA(A)=f(WAA+bA),WA∈Rd′×dA,bA∈Rd′

(3) Tproj=PT(T)=f(WTT+bT),WT∈Rd′×dT,bT∈Rd′.

Here, WV, WA, and WT are learnable weight matrices that project the features into a shared space of dimension d′. The bias terms bV, bA, and bT ensure flexibility in projection. The activation function f (e.g., ReLU) introduces non-linearity, enabling the model to capture complex relationships within each modality. By projecting all modalities into a shared space of the same dimension d′, we ensure dimension compatibility and facilitate meaningful cross-modal interactions.

Cross-modal projection and dimension compatibility

To further enhance the multi-modal feature fusion, we introduce cross-modal projections, where each modality is mapped into spaces aligned with other modalities. Specifically, video features are projected into both audio and text feature spaces, while audio and text features are projected into a video space:

(4) VprojA=PVA(V)=f(WVAV+bVA),WVA∈Rd′×dV,bVA∈Rd′

(5) VprojT=PVT(V)=f(WVTV+bVT),WVT∈Rd′×dV,bVT∈Rd′

(6) AprojV=PAV(A)=f(WAVA+bAV),WAV∈Rd′×dA,bAV∈Rd′

(7) TprojV=PTV(T)=f(WTVT+bTV),WTV∈Rd′×dT,bTV∈Rd′.

Each projection matrix WXY ensures that all projected features share the same dimensionality d′, enabling direct comparison and fusion. This alignment of feature spaces is critical to ensure compatible and meaningful interactions between modalities.

Feature fusion

Following cross-modal projections, we concatenate all aligned feature representations to construct a unified multimodal embedding:

(8) Fcombined=[VprojA,VprojT,AprojV,TprojV].

This concatenation-based fusion approach is applied for the following reasons: All features are projected into a shared latent space of dimension d, ensuring compatibility and meaningful interactions across different modalities.

Unlike averaging or attention-based approaches, concatenation preserves distinct modality characteristics, preventing the loss of critical unimodal information.

By maintaining separate feature streams, the model can learn complex inter-modal dependencies in subsequent layers without enforcing any predefined relationships between modalities.

Since the final fused feature vector Fcombined resides in R4d, it is then processed by a classifier:

(9) y^=Classifier(Fcombined).

The classifier, typically comprising fully connected layers, learns a discriminative multimodal representation that optimally combines the projected features to make accurate predictions.

Evaluation setup

To evaluate our model for multimodal emotion detection in Urdu, we created a specialized corpus called the UMED corpus, which stands for Urdu Multimodal Emotion Detection corpus. It is a collection of annotated video, audio, and text designed to cover a variety of emotions. The creation process was divided into several steps: data collection, annotation, and validation. In the following sections, we will discuss each step in detail, including the tools and technologies used.

Corpus collection

In all NLP approaches, benchmark corpora are vital instruments that derive methodologies and tools. This research is specifically dedicated to contributing to the field of multimodal emotion detection for the Urdu language. Regrettably, a standard corpus for the Urdu language is not available for multimodal emotion detection, as is the case for other languages. One of the primary goals of this research is to fill the gap by providing a custom-made benchmark corpus in the Urdu language, specially designed for multimodal emotion-based analysis. The initial challenge faced in creating this corpus was finding relevant data sources. Primary sources were selected from Facebook, DramaOnline, YouTube, and DailyMotion. The processes used to acquire this raw data were both automatic and manual. The diversity of the corpus was ensured by obtaining data from multiple regions. This meticulous data collection method aims to present a corpus that covers a wide range of emotional expressions in Urdu reflecting different situations and domains.

Real-time emotion annotation

As shown in Fig. 6, we have developed a web app to enable annotations. The annotation software we built was meticulously designed to ensure usability and facilitate an intuitive workflow. Annotators were met with a clean interface that provided elegant functionality and helped them with easy navigation and efficient annotation. The interface had a video playback window placed next to the annotation panel. Consequently, an annotator could watch the video and annotate each sentence in real time. From the annotation panel, annotators could see a row of buttons assigned to the respective emotion classes. Each class was labeled correctly so that it could be easily recognized. The annotators progressed through the video by clicking the appropriate button to assign the relevant emotion class to the sentence. Whenever the button is clicked, the app automatically crops the video fragment annotated by the sentence and saves it to a specified folder with the corresponding label.

Figure 6 Annotation tool interface allows users to upload a video and select an emotion label for the displayed video frame.

Data annotation

In this work, we intend to develop an automated system for the Urdu language, specifically focusing on multimodal emotion detection. To achieve this goal, we required an annotated corpus with labeled emotions. For this purpose, we used the five fundamental emotions: happy, sad, anger, love, and neutral. The development of an annotated corpus starts once raw data has been collected. Then, the expert annotators filter it using the five emotions discussed earlier. The annotators comprised of five experts. Annotation rules and guidelines for problematic and often occurring labeling ambiguities were provided to annotators with multiple samples in advance. In addition, we also prepared a training section, and weekly seminars were held so that the participants could have interactive discussions to work out annotation problems, thereby providing us with informed feedback on the whole annotation process as shown in Fig. 7. A set of guidelines were provided to the annotators which they were strictly required to abide by. 1. Assign one of the five class labels to each example.

2. If an example is not assigned to any predefined emotion class, mark it as neutral.

3. If an example is part of several emotions, it allocates a class of emotions closest to that example based on the context.

4. The annotation of emotions is limited to the nature of tagged sentences only.

Figure 7 Annotation process for UMED corpus: this diagram demonstrates the process which follows first extracting raw data and uploading the video into the annotation tool then annotating according to rules for addition to the final corpus.

After completing the data annotation, we analyzed the annotated data and found that the corpus was unbalanced across the emotional classes. For this purpose, YouTube videos were used to collect supplemental data to accompany the existing corpus to ensure the balance and comprehensibility of our dataset for training and evaluation purposes. After completing the annotation process, we converted the videos into audio and then transcribed the audio into text. This process enables the formation of a corpus adding visual, text, and speech, for widespread emotional expression in the Urdu language across different modalities. The main struggle with working on multimodal data involves properly synchronizing different modalities comprising text, audio, and video. The dataset structure relies on a CSV file which provides synchronized links between videos, speech, and text. Such alignment methods between all modalities lead to better model training efficiency.

Corpus characteristics

After the annotation, we performed a statistical analysis of the corpus to explore the characteristics of emotional expressions. The fundamental statistics of the corpus are presented in the Table 2. The Table 2 provides the total number of raw videos which vary in length and contain multiple sentences per video. The raw corpus consists of over 125 h of video data. Using our annotation tool, annotators assigned emotion labels to each sentence in the video while watching them. The average duration of annotated video segments is 8 s, with the maximum segment length reaching 15 s. Following the annotation process, the finalized corpus comprises approximately 17 h of annotated data, comprising 8,278 annotated video segments. The Table 2 also presents the total number of examples for each emotion class. Figure 8 further illustrates the distribution of emotion categories by the time duration of each class, corresponding to the different classes within the corpus.

Table 2 Overall corpus statistics.

Whole corpus statistics	
Total number of raw videos	450	
Total time of raw corpus	125 h	
Annotated corpus statistics	
Total number of annotated videos	8,278	
Total time of annotated corpus	17 h	
Emotion category statistics	
Total number of emotions	5	
Count of happy examples	1,771	
Count of sad examples	1,624	
Count of angry examples	2,068	
Count of love examples	1,135	
Count of neutral examples	1,680	

Figure 8 Time wise frequency of corpus.

Validation of the corpus

The corpus validation process plays a critical role in this study. During the annotation phase, videos were assigned to annotators, and they labeled each sentence with one of five emotional categories. We employed five annotators for this task. Each video was assigned to at least two distinct annotators from the group of five. As discussed in the previous section, the video was uploaded to our annotation tool for annotation. After the initial annotation by two annotators, the remaining three annotators were tasked with validating the labels. The annotation was considered valid if all three agreed on the same emotion label. Otherwise, the label was revised after a discussion with the annotation team. In cases where no consensus is reached after three annotations, the corresponding entry is removed from the corpus as shown in the Fig. 9. Additionally, we measured the inter-annotator agreement using Cohen’s Kappa score, which was found to be 0.80, indicating a strong level of agreement. Finally, we analyzed our final corpus, which contained 8,278 video segments, to ensure gender balance. These videos feature individuals from various age groups, ranging from 13 to 80 years old, representing a wide demographic. All videos adhere to our specific criteria:

Figure 9 Flow of validation process.

1. Video must contain one speaker.

2. There must be no background noise or music in the video and the speaker’s voice should be clear.

3. The environment should be properly lit, which makes the speaker visible.

4. The speaker must be speaking in the Urdu language.

Implementation of UMEDNet

The UMEDNet architecture consists of several key components. First, the input modalities are projected with learnable matrices into the shared space of size d′ before being processed with activation functions. This step ensures dimensional compatibility and aligns feature distributions across different modalities. Each single modality undergoes an additional projection step to match the dimension space of connected modalities. The multimodal alignment improves when video features are projected to audio and text spaces simultaneously, with the same process for audio and video features. A unified multimodal embedding integrates projected features after integrating specific attributes of each modality while making it possible for the model to learn complex cross-modal relationships. The final step includes applying fully connected layers with dropout regularization onto the merged multimodal embedding to complete emotion classification using a softmax layer.

Using the UMEDNet model users can detect emotions in the Urdu language from video, audio, and textual sources. The modality components operate alone before entering a shared latent space enabling proper alignment for effective fusion. The Wav2Vec2 model obtains audio embeddings by applying its Urdu-specific adaptation to the pre-trained multilingual dataset through fine-tuning. Assessing text embeddings requires XLM-R which underwent successful fine-tuning as it releases contextual understanding of Urdu text relations. Video features are obtained using MTCNN for face detection and FaceNet for facial feature extraction, which helps capture visual cues such as facial expressions and movements. A uniform dimension was maintained via common latent space for embeddings achieved through learnable transformation matrices during feature extraction. The fused multimodal embeddings proceed to a classification network for emotion prediction.

Extensive hyperparameter tuning was conducted using a validation set to achieve optimal performance. The study of learning rate between {1e−3, 5e−4, 1e−4, 5e−5} showed that 1e−3 delivered the optimal balance of speed and stability during convergence. A batch size of 32 delivered the optimal results from the examined options 16, 32, and 64. Weight decay values tested {1e−4, 1e−5, 1e−6} the 1e−5 value became the optimal choice because it managed to limit overfitting without compromising generalization capabilities. From different dropout rates {0.1, 0.2, 0.3, 0.5} the value of 0.2 exhibited the most effective regularization effect. The analysis included multiple fusion techniques such as concatenation and averaging but concatenation was chosen because it maintained original modality information and enabled efficient cross-modal relationship learning. The final selected hyperparameter values were chosen because they resulted in the most optimal F1 score on the validation set.

The training process for UMEDNet was conducted as a multi-class classification via cross-entropy loss optimization. A learning rate of 0.001 coupled with weight decay at 1e−5 function within the Adam optimizer during which the optimization happens. The training runs for 100 epochs but includes an early stopping technique to monitor validation loss because of its capacity to stop overfitting. Robustness in the model is enhanced through multiple implemented regularization and generalization approaches. The model integrates Dropout layers with a rate of 0.2 across its entire structure for overfitting prevention. Batch normalization serves to stabilize training while speeding up the convergence process of the model. Standard evaluation metrics such as accuracy alongside precision, recall, and F1 score are applied to a separate test set for model evaluation.

The complete model framework from feature extraction to projection fusion and classification operates under PyTorch with graphics processing unit (GPU). The Hugging Face Transformers library enables fine-tuning for Wav2Vec2 and XLM-Roberta whereas MTCNN and FaceNet implementation originates from the facenet-pytorch library. This integrated framework provides end-to-end multimodal emotion detection for the Urdu language.

Experimentation and result analysis

We evaluated UMEDNet’s performance by using our corpus and applying several comparing techniques. Having enough amount of training data while leaving validation and test sets for parameter tuning and final evaluation, the dataset was split into 70% training, 15% validation, and 15% testing. Hyperparameter selection and early stopping to prevent overfitting were done using the validation set. We trained and tested the model several times using 5-fold cross-validation to increase reliability. This helped reduce overfitting and made it more generalizable. A stable assessment was carried out for performance metrics by averaging across folds. We carefully split the corpus and minimized the data leakage to a robust evaluation of UMEDNet in multimodal emotion detection.

We utilized Huggingface, PyTorch, pandas, Scikit-Learn, Natural Language Toolkit (NLTK), and the transformers library to test the model’s accuracy and performance. Personal computers, Kaggle, and Google Colab were used for the experiments. The evaluation of results was carried out through accuracy, F1 score, precision, and recall. Evaluation details are discussed below.

Umednet emotion detection results: performance overview

The proposed UMEDNet achieved an overall accuracy of 85.27%, indicating strong performance in classifying emotions using multimodal data consisting of video, audio, and text. Additionally, the precision, recall, and F1-score are consistently high at 85.40%, 85.27%, and 85.29% respectively, demonstrating that the model effectively balances precision and recall, as shown in Table 3.

Table 3 UMEDNet performance for emotion prediction.

Class	Precision	Recall	F1-score	Support	
0	88.89	83.58	86.15	201	
1	80.60	87.10	83.72	186	
2	87.16	87.16	87.16	109	
3	84.44	83.98	84.21	181	
4	86.58	85.43	86.00	151	
Accuracy		85.27		828	
Macro avg	85.53	85.45	85.45	828	
Weighted avg	85.40	85.27	85.29	828	

The training and validation accuracy Fig. 10 illustrates the performance of the UMEDNet model throughout training. The training accuracy steadily increases and in the later epochs, it stabilizes around 88%, indicating that the model is learning effectively without showing signs of over-fitting. Meanwhile, the validation accuracy plateaus around 84–85%, aligning closely with the reported overall accuracy. This slight gap between training and validation accuracy suggests that the model generalizes well to unseen data, maintaining strong performance across different datasets.

Figure 10 UMEDNet: training and validation accuracy over the epochs.

The confusion matrix in Fig. 11 provides deeper insights into the model’s performance across specific emotion categories. Particularly, the model excels in classifying emotions like anger and happy, achieving a high number of correct predictions. Neutral and sad classes also show strong prediction accuracy, but there are some instances of misclassification. For example, the love class exhibits occasional overlap with other categories, reflecting the challenge of distinguishing between emotions with subtle differences. This confusion between similar emotions, such as between neutral and sad, may stem from inherent ambiguities in the dataset, where certain emotional features overlap. Nonetheless, the overall distribution of correct predictions remains strong, underscoring the robustness of UMEDNet in predicting emotions across various categories. Thus, UMEDNet demonstrates effective multimodal emotion detection for Urdu, achieving competitive accuracy and classification consistency across diverse sets of emotions.

Figure 11 Confusion matrix of results from UMEDNet.

Umednet: performance comparison between projected and non-projected features

We analyze the performance of UMEDNet with and without feature projection. The utilization of feature projection plays a critical role in aligning multimodal features into a common space, facilitating a more effective fusion of video, audio, and text data. This comparison aims to demonstrate the impact of feature projection on the model’s ability to predict emotions accurately.

The results presented in Table 4 indicate that feature projection significantly improves the model’s performance across all key evaluation metrics. When projection is applied to the video, audio, and text features, UMEDNet achieves an accuracy of 85.27%, as compared to 82% without projection. Similarly, with projection, the precision is 85.40% and recall is 85.27%, but these metrics drop to 82.0% without using projection. The F1-score also shows an improvement with projection, with a slight increase from 81.90% to 85.29%. By aligning the feature spaces of different modalities, UMEDNet better captures the relationships between them, resulting in more accurate and balanced predictions. The model struggles to exploit the complementary information the different modalities provide without feature projection, reducing its overall performance.

Table 4 Performance comparison of model with variations.

Models	Accuracy	Precision	Recall	F1-score	
UMEDNet with projection	85.27	85.40	85.27	85.29	
UMEDNet without projection	82.0	82.10	82.0	81.90	
RNN-UMEDNet	77.66	77.69	77.66	77.63	

Evaluating umednet: impact of model alterations on performance

We evaluated the performance of the proposed UMEDNet model in comparison to an altered version, referred to as RNN-UMEDNet. The key difference lies in replacing the transformer used for video feature extraction with a recurrent neural network (RNN). This comparison highlights the impact of this architectural change on the model’s ability to predict emotions from multimodal data.

As shown in Table 4, the results indicate a clear performance gap between the two approaches. The proposed UMEDNet, utilizing a transformer for video processing, outperforms the RNN-UMEDNet in all evaluation metrics. UMEDNet achieves an accuracy of 85.27%, whereas RNN-UMEDNet achieves only 77.66%. Similarly, precision, recall, and F1-score drop from 85.40%, 85.27%, and 85.29%, respectively, to 77% across the board for the RNN-UMEDNet model. This decline in performance suggests that the transformer-based model is more effective at capturing the temporal dependencies and nuanced features in video data as compared to the RNN-based model.

Evaluating umednet across modality combinations

The results in Table 5 show how the UMEDNet model performs with different combinations of modalities: video, speech, and text. Each modality pair contributes distinct information, influencing the model’s ability to predict emotions effectively.

Table 5 Comparison of model performance using different modality combinations for emotion detection.

Modalities	Accuracy	Precision	Recall	F1-score	
Speech + Text	62.20	60.50	62.20	61.78	
Video + Speech	79.50	80.75	79.50	79.32	
Video + Text	82.0	82.20	82.0	81.85	
Video + Speech + Text	85.27	85.40	85.27	85.29	

When the model uses an input of text and speech only, the accuracy derived is the lowest being 62.20%. This result indicates that, though both speech and text are used for assessing verbal and linguistic cues, they are insufficient to address nonverbal expressions of emotion. When both, the video and the speech input are used, the model overall accuracy is 79.50%. Therefore, video features that capture a person’s facial conduct will improve the model’s accuracy. Another fact about emotions is that while identifying them, one has to consider visuals that can contain other information apart from audio and text. Adding the results of the video analysis and text increases the accuracy to 82%. This improvement proves that as important as text is in delivering semantic content, it is when combined with a video that a deeper feel of the emotional content through facial expressions and visuals is gained. Lastly, using all three modalities, video, speech, and text, the model yields the highest accuracy of 85.27%. This only underscores the use of multiple sources of emotion detection, all of which feed off each other. The video has facial expressions, the speech affects tone and rhythm, and the text carries semantic meaning, all of which are crucial for conveying the emotions of someone.

Discussion

The development of models and corpora for Urdu presents multiple challenges because the low-resource Urdu language lacks extensive labeled datasets. The UMED corpus originated from various sources through a process that included annotator training, distinct guideline development, and the creation of a dedicated annotation tool for data quality management. The strict preprocessing techniques allowed us to resolve additional issues caused by dialect variations and informal styles. Transfer learning through fine-tuning Wav2Vec2 and XLM-Roberta models solved the problems associated with low-resource languages. The foundation of adaptation to Urdu benefitted from these pre-trained models which initially processed high-resource languages. Our system includes a multimodal framework that combines different features to boost the performance of emotion prediction. The combination of these strategies together fills the resource deficit and boosts the Urdu emotion recognition model’s capability to generalize effectively.

Additionally, to evaluate the roles of each component in the proposed UMEDNet, we compared its performance when feature projection was included and when it was removed. The results here suggest the improvement of feature projection boosts the model’s accuracy from 82% to 85.27%. This improvement indicates that aligning the feature space of video, audio, and text modalities helps in better fusion of information and results in better prediction of emotion. The comparative analysis of UMEDNet and RNN-UMEDNet displays the benefits of transformers for video analysis. UMEDNet is statistically significantly better than the RNN-based architecture throughout the evaluation process with 85.27% accuracy as compared with RNN-UMEDNet’s 77.66% accuracy. This performance difference further establishes the effectiveness of transformer-based architectures in modeling temporal dependencies and finer details within the video data, which supports the architectural decisions made for UMEDNet.

Moreover, comparing UMEDNet for all the modality combinations offers better perspectives on the significance of multimodal fusion. The studies point out that although features like speech and text bear useful information, they are ineffective when used separately to detect emotions. However, the performance enhancement is much more prominent when different modalities are combined especially those with video modality. For instance, using video in conjunction with speech or text leads to a notable increase in accuracy, with the highest performance achieved when all three modalities are utilized together.

Conclusion

In this research, we introduced UMEDNet, a new multi-modal emotion prediction model developed particularly for the Urdu language. Due to the lack of emotion detection models for low-resource languages, the presented approach fills an important gap as a feasible solution for multi-modal emotion classification. We also developed the UMED Corpus which contains video, speech, and text in Urdu, a critical resource for advancing research in this domain. As learned from our experimental results, UMEDNet proved effective in capturing and classifying emotional states accurately through the proposed multimodal fusion approach. Transformer and FFN were then used to process the features and project all the features into a shared latent space so that the multiple modalities could be aligned and fused appropriately in UMEDNet. This feature projection process was useful for enhancing the performance of emotion recognition since it enhanced the ability of the model in terms of interpretation and integrating the emotional signals coming from the various modalities.

Overall, UMEDNet presents a strong and stable approach for emotion detection in the Urdu language that can be very effective in research of real-world scenarios. Besides aiding this model, the development of the UMED corpus can also be important to other research for multimodal emotion detection. Future work could be conducted to gather more data, enhance multimodal fusion approaches, and identify the possible application areas such as mental health monitoring, emotionally-aware conversational agents, and social media analysis.

We want to acknowledge the creation of the UMED corpus, which we developed specifically for this research. The entire process of corpus creation is discussed in the evaluation setup section.

Additional Information and Declarations

Competing Interests

The authors declare that they have no competing interests.

Author Contributions

Adil Majeed conceived and designed the experiments, performed the experiments, analyzed the data, performed the computation work, prepared figures and/or tables, authored or reviewed drafts of the article, and approved the final draft.

Hasan Mujtaba conceived and designed the experiments, analyzed the data, authored or reviewed drafts of the article, and approved the final draft.

Data Availability

The following information was supplied regarding data availability:

The code is available at GitHub and Zenodo:

- https://github.com/AdilMajeed/UMED/tree/main.

- AdilMajeed. (2025). AdilMajeed/UMED: UMEDNet (UMED). Zenodo. https://doi.org/10.5281/zenodo.14925456.

The data is available at Zenodo: National University of Computer and Emerging Sciences. (2024). UMED Corpus [Data set]. Zenodo. https://doi.org/10.5281/zenodo.13988610.

The complete dataset is available at Zenodo: AdilMajeed. (2025). AdilMajeed/UMED: UMEDNet. Zenodo. https://doi.org/10.5281/zenodo.15183245.

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
