# Peer review of "UMEDNet: a multimodal approach for emotion detection in the Urdu language"

_PeerJ Computer Science, doi:10.7717/peerj-cs.2861_

## Round 0.1 · original submission · Major Revisions

· Academic Editor

Major Revisions

The paper presents an original contribution to multimodal emotion detection in the Urdu language. However, as the reviewers suggested, some key areas need improvement to enhance clarity and scientific rigour. Please respond to all their comments to strengthen the content and its impact. In particular, improve the part about the dataset and methodology with more details and better justify the mathematical framework/model. Reviewers expressed concerns about the identical values for accuracy, precision, recall, and F1 score across models and ask for a more comprehensive literature review and a comparative analysis against other state-of-the-art models.

·

Basic reporting

The authors presented a multimodal approach for emotion detection in the Urdu language. The paper demonstrates originality and scientific rigour.

Experimental design

Please explain the dataset in more detail specifically focusing on the dataset characteristics and preprocessing challenges with multimodal data; synchronization of text, audio, and video.

Validity of the findings

The results are explained well. However, I will suggest highlighting potential limitations of the results, such as biases in data or constraints in generalizability for further improvement.

Reviewer 2 ·

Basic reporting

Please see Additional comments

Experimental design

Please see Additional comments

Validity of the findings

Please see Additional comments

Additional comments

1. The authors are suggested to provide a comparative analysis of UMEDNet's performance against existing state-of-the-art multimodal emotion detection models.

2. While MTCNN, FaceNet, Wav2Vec2, and XLM-Roberta are mentioned as feature extraction techniques, the paper does not adequately justify why these methods were chosen over others. For example, newer models like DINOv2 (for vision) or HuBERT (for speech) might provide enhanced performance.

3. Although the study focuses on Urdu as a low-resource language, it does not discuss the specific challenges encountered in developing models or corpora for such languages, nor does it provide strategies for addressing these challenges (e.g., transfer learning from high-resource languages).

4. The paper briefly mentions overall accuracy, precision, recall, and F1 scores, but it lacks detailed performance metrics per modality and emotion category.

5. The author has missed multiple important and closely related articles in the literature review.
Towards enhanced identification of emotion from resource-constrained language through a novel multilingual BERT approach
Emotion detection using convolutional neural network and long short-term memory: A deep multimodal framework
A machine learning-based investigation utilizing the in-text features for the identification of dominant emotion in an email
Identifying dominant emotional state using handwriting and drawing samples by fusing features

Cite this review as

·

Basic reporting

This paper presents UMEDNet, a multi-modal emotion prediction model for Urdu, and the UMED Corpus, a key resource for emotion classification in low-resource languages. The model effectively classifies emotions using video, speech, and text data.

1. In the "Related Works" section, it would be beneficial to provide a more comprehensive discussion of the challenges faced by low-resource languages in both emotion prediction and multi-modal models. Specifically, the paper could elaborate on the limitations posed by the scarcity of publicly available datasets and pre-trained models, which are major obstacles in these fields. It would also be valuable to explore existing solutions, such as cross-lingual transfer learning, and offer a comparison between these approaches and multi-modal studies. Additionally, including relevant literature on similar studies conducted in languages such as Tamil or Haitian Creole would further strengthen the context and breadth of the research.
2. In the "Corpus Collection" section, further elaboration on the data quality and diversity would enhance the comprehensiveness of the discussion. Specifically, it would be valuable to address the balance between formal and informal language (or written vs. spoken language) in the collected data. Additionally, the inclusion of demographic factors, such as age, gender, and other relevant variables, would provide a more nuanced understanding of the corpus' representativeness. Furthermore, a discussion on the overall size of the dataset would be beneficial, as this impacts the generalizability and robustness of the findings.
3. In the "Data Annotation" section, it would be useful to outline the measures taken to ensure the quality and reliability of the annotated data. Specifically, clarifying the procedures followed to guarantee that the annotated video meets high standards, often referred to as "gold" annotations, would strengthen the validity of the dataset. This could include details on inter-annotator agreements, quality control processes, and any validation methods employed.
4. Some minor typos and grammatical suggestions for improvement:
- Use hyphenated “self-supervised” in line 98 “Self Supervised Learning”.
- Around line 100: “achieves” in “jointly fine-tuning ”BERT-like” SSL designs…”
- In the second row of Table 1, the “Model” column seems to be missing a blank space between “SSLmodels”.
- Around line 306: “One of the things we have set out to do through this research 306 is fill this gap by providing a custom-made benchmark corpus in the Urdu language, especially designed 307 for multimodal emotion-based analysis. ” could use a slight revision to “is to fill…”. Also, change "especially designed" to "specifically designed" to make the description more precise.
- In line 318, hyphenated “real-time” is usually used as an adjective to modify nouns. Therefore, “real time” would be more accurate in this case.
- In line 338, either continue using “allocates” with a newly-added subject “the system”, or change to “allocate”.
- In Table 2, change “Total numbers” to “Total number”.
- In line 364, there appears to be an extra blank space after “...previous section .”.

Experimental design

Selecting a pre-trained XLM-Roberta for text representation might introduce bias for overrepresented languages, which could effectively affect the model performance in low-resource multi-modal tasks. while the paper does not need to address this issue, it would be helpful to include it in the discussion.

In addition, it would be helpful to mention the exact model architecture for both wav2vec2 and XLM-Roberta used in this paper. For example, does the context length window change, or how many attention heads are used(if different from the default).

Validity of the findings

The findings of the paper are valid as it is supported by experiments conducted on collected datasets. Please address the comments mentioned in the previous sections.

Additional comments

Thanks for the opportunity to review the manuscript. Please address the comment above.

Reviewer 4 ·

Basic reporting

Strengths:

-Well-structured paper following standard scientific format
- Clear presentation of methodology
-Comprehensive literature review establishing context

Areas for Improvement:
-Some grammatical errors and awkward phrasing need correction: For example:
--- Replace
" Interpersonal relationships, knowledge, insight, and perception,
31 among other things, depend on emotion Cevher et al. (2019)Dadebayev et al. (2022)"
---By
"Interpersonal relationships, knowledge, insight, and perception, among other things, depend on emotion (Cevher et al., 2019; Dadebayev et al., 2022)"

--- Replace
"In our model, we employ projection functions to project the features of various modalities"
--- By
"Our model employs projection functions to map features from various modalities"

-Several acronyms are used without first defining them (e.g., MTCNN, FFN)
-Figures 2-4 could benefit from more detailed captions explaining the components
- Results section should be totally reviewed and corrected given the accuracy, precision and recall results numbers.

Experimental design

*Key strengths of the experimental design in UMEDNet:

-Corpus Development


Systematic data collection from diverse sources
Rigorous annotation process with 5 expert annotators
Strong inter-annotator agreement (Cohen's Kappa 0.80)
Clear validation criteria for video quality and content


*Mathematical Framework Issues:

-Equations (1-3):
Basic projection functions lack mathematical rigor
No explanation of dimension compatibility between modalities

-Feature Fusion:
Equation (8) concatenation approach is simplistic
No theoretical justification for projection space alignment
Missing mathematical proof of feature space compatibility

*Experimental Design Concerns:
-Dataset Splitting:
70/15/15 split needs justification
No cross-validation methodology
Potential data leakage not addressed

-Hyperparameter tuning and model architecture:
Tuning should be covered , "Implementation of UMEDNet" section should be extended and talk more about the implementation.

Validity of the findings

The results section should be totally corrected

- In line 402 and table 3, you state that "Additionally, the precision, recall, and F1-score are consistently high at 86%, 86%, and 85% respectively, demonstrating that the model effectively balances precision and recall, as shown in Table 3." if precision and recall are 86%, also F1 = 2Precision*Recall/(Precision+Recall) should be equal to 86% not 85%. I know you most likely rounded these values, it will look better if you use 2 digits precision.

Also the fact that your accuracy=precision=recall, is suspicious.

- Same comment about table 4 and table 5, your UMEDNet with Projection , UMEDNet without Projection , RNN-UMEDNet results shows that accuracy=precision=recall=F1 score

Having identical values for accuracy, precision, and recall is unusual and raises concerns:
-Statistical Improbability:
These metrics typically vary due to different error types
Perfect balance across all metrics is extremely rare in real-world scenarios


Technical Issues:
Could indicate problems with evaluation methodology
May suggest issues with the test set distribution
Possible implementation errors in metric calculations


Reporting Issues:
Might be rounded to same value (actual numbers could differ)
Could be oversimplified reporting of results
Raw confusion matrix numbers would help verify these metrics

The authors should:

Provide unrounded metric values
Include per-class metrics
Explain this unusual statistical outcome

This identical performance across metrics reduces confidence in the evaluation methodology and should be addressed in revision.

Cite this review as

---

## Round 0.2 · accepted · Accept

· Academic Editor

Accept

The authors have adequately addressed all of the reviewers' comments. I confirm that this manuscript is ready for publication.

·

Basic reporting

The manuscript is now clear and well structured. Literature review is sufficient, and results are well explained

Experimental design

The research work is under the scope of the journal. The authors have improved the methodology, and sufficient investigation is performed to fill the identified gap.

Validity of the findings

Improved

Reviewer 2 ·

Basic reporting

The revised paper is in a better shape.

Experimental design

The revised paper is in a better shape.

Validity of the findings

The revised paper is in a better shape.

Additional comments

The revised paper is in a better shape.

Cite this review as

·

Basic reporting

The authors have successfully addressed the comments raised in my previous review.

Experimental design

Same as above.

Validity of the findings

The paper's findings are valid as they are supported by robust experiments conducted on the UMED Corpus.